# Is Consultation-Liaison Psychiatry ‘Getting Old’? How Psychiatry Referrals in the General Hospital Have Changed over 20 Years

**DOI:** 10.3390/ijerph17207389

**Published:** 2020-10-10

**Authors:** Silvia Ferrari, Giorgio Mattei, Mattia Marchi, Gian Maria Galeazzi, Luca Pingani

**Affiliations:** 1Department of Biomedical, Metabolic Sciences and Neurosciences, University of Modena and Reggio Emilia, Via del Pozzo 71, 41124 Modena, Italy; silvia.ferrari@unimore.it (S.F.); giorgio.mattei@unimore.it (G.M.); mattia-marchi@libero.it (M.M.); gianmaria.galeazzi@unimore.it (G.M.G.); 2Department of Economics “Marco Biagi”, University of Modena and Reggio Emilia, Via Jacopo Berengario 51, 41121 Modena, Italy; 3Department of Health Professions, Azienda USL–IRCCS di Reggio Emilia, Via Amendola 2, 42122 Reggio Emilia, Italy

**Keywords:** consultation-liaison psychiatry, elderly, psycho-geriatrics, general hospital

## Abstract

There is an ever-growing awareness of the health-related special needs of older patients, and Consultation-Liaison Psychiatry Services (CLPS) are significantly involved in providing such age-friendly hospital care. CLPS perform psychiatric assessment for hospitalized patients with suspected medical-psychiatric comorbidity and support ward teams in a bio-psycho-social oriented care management. Changes in features of the population referred to a CLPS over a 20-year course were analysed and discussed, especially comparing older and younger referred subjects. Epidemiological and clinical data from all first psychiatric consultations carried out at the Modena (North of Italy) University Hospital CLPS in the period 2000–2019 (*N* = 19,278) were included; two groups of consultations were created according to the age of patients: OV65 (consultations for patients older than 64 years) and NONOV65 (all the rest of consultations). Consultations for OV65 were about 38.9% of the total assessments performed, with an average of approximately 375 per year, vs. the 589 performed for NOV65. The number of referrals for older patients significantly increased over the 20 years. The mean age and the male/female ratio of the sample changed significantly across the years in the whole sample as well as both among OV65 and NOV65. Urgent referrals were more frequent among NOV65 and the rate between urgent/non urgent referrals changed differently in the two subgroups. The analysis outlined recurring patterns that should guide future clinical, training and research activities.

## 1. Introduction

Consultation-liaison psychiatry (CLP) is a branch of psychiatry aimed at dealing with the complex interactions of ‘multi-morbidity’ [1], including its psycho-social causes and effects. Born with a strong operative accent, i.e., how to perform psychiatric consultations on patients admitted to the general hospital (GH) or in other contexts of health care (“consultation”), and how to support and favour communication among health care professionals (“liaison”), it has moved to be the operationalized arm of psychosomatic medicine and of the bio-psycho-social paradigm promoted by George Engel and many other researchers and clinicians over the second half of 20th century [2,3,4].

CLP everyday activities in the GH most commonly consist in performing diagnostic assessments and activate treatment plans when medical-psychiatry comorbidity may be present, including adjustment disorders, anxiety, self-harm behaviours and suicidality, delirium, medically unexplained physical symptoms, eating disorders, alcohol and substance-related disorders, and so forth. Prevention of work-related stress and burn-out syndrome of health professionals as individuals or teams is another typical target of a CLP Service (CLPS) [3]. Strong interaction with related health and social agencies inside and outside the GH (i.e., general practice or social services) is one of the core principles of CLP, to achieve better integration and coordination of care, rationalization of health care resources and reduction of excessive medical sectorialization [5,6].

Research has confirmed how effective CLP clinical actions may help improving outcome indicators of health care, i.e., long-term prognosis of medical conditions, adherence to care plans, quality of life and disability of patients, length of hospital stay, health direct and indirect costs [7,8,9,10]. 

Older patients represent the vast majority of patients admitted to the GH, especially in western countries, with rates ranging between 40 and 65% [11,12]. Referrals for in-patients aged more than 65 have increased consistently, e.g., raising from 0.7 to 2.89% over a 30-year period in an UK CLPS [13]. Old age is a significant risk factor for many psychiatric conditions that may complicate the course and management of multimorbidity, such as depressive and anxiety disorders and neurocognitive disorders [11,14]: up to 40% of geriatric patients may present concomitant psychiatric complaints [15]. Complex bio-psycho-social health care needs impact massively on health outcomes of older patients [16], but may still be neglected, mostly due to delayed diagnosis, inadequate use of psychotropic medications and other therapeutic tools, fear of stigma and limited integration of care. Referrals for psychiatric consultation is performed less frequently for old age patients than other age ranges [15].

In the “*Global Age-Friendly Cities: A Guide*” published by the WHO, older people from 33 different countries were asked to discuss positive and negative aspects of their living situations; when reporting on health-related issues, disease prevention and health promotion were cited as the most relevant expectations, whereas lack of coordination among services, causing complications and reducing effectiveness, was quoted as a common problem [17]. Indeed, taking care of older patients both inside and outside the hospital setting requires specific attention combined to dedicated skills. For example, the rate and severity of multimorbidity increases with the age of patients, and, consistently, efforts in providing coordinated and integrated care should be made. This is known to improve health performances in terms of prognosis and quality of life, as well as reducing and rationalising health costs [18]. Interestingly, the same topics are among the most important targets of CLP, particularly by means of a close interaction with community health services [19,20]. The solid holistic tradition of psychosomatic medicine inspires all CLP’s clinical, training and research actions, aiming at making the statement “the patient at the centre” not just an empty metaphor but a real effort towards autonomy and active involvement of patients in their care action plans [21,22].

Aim of this paper was describing changes of socio-demographic and clinical characteristics in the population referred to a hospital-based CLPS over a 20-year (2000–2019) lag-time. Specifically, the variables were examined as referred to the whole sample of older people (aged 65 or more—OV65) and in comparison, to the rest of the population assessed by the service (aged between 18 and 64—NOV65).

## 2. Methodology

The Modena GH is a 611-bed university hospital, belonging to the Regional Health System (Regione Emilia-Romagna); it is located in the central city area of Modena, a middle town in the North of Italy, and its catchment area is of about 200 thousand inhabitants. It includes an Accident & Emergency (A&E) unit. The Modena CLPS was instituted in 1989 by Prof. Marco Rigatelli and administratively belongs to an operative unit of Adult Mental Health Care, also including a 12-bed psychiatric rehabilitation residential facility and a community mental health service. The CLPS is staffed by three half-time psychiatrists and three to four full-time residents in psychiatry; residents stay for 12 months at the CLPS, during the intermediate-final stages of residency (moderate-to-high level of clinical autonomy according to Italian regulations for resident training). A varying number of final-year medical students and volunteer medical doctors regularly take part to clinical and research activities. PhD students at pertinent Doctorate Schools of the University of Modena & Reggio Emilia also collaborate to research activities. The CLPS does not staff other professionals than medical doctors (e.g., nurses, clinical psychologists, social workers) [23]. 

The CLPS provides routine (non-urgent) and urgent psychiatric referrals to all the wards of the GH, excluded the paediatric ward and included the A&E unit. Patients referred are mostly in-patients, but protocols of collaboration with a limited number of out-patient clinics are available. The CLPS is open on working days from 8 am to 3 pm and on Saturdays from 8 am to 1 pm. In the remaining hours and during festivities, only for urgent referrals, the 24/7 on-call general psychiatrist (operating in a different hospital of the town) is contacted. 

Request forms for psychiatric consultations are sent electronically through the IT system. The form specifically differs from generic consulting request forms for all other medical specialties and is semi-structured for psychiatric referrals, to help clinicians in the formulation of a more defined clinical question [3,24] and to provide consultants with relevant information; particularly, a list of possible pre-defined reasons for referrals and diagnostic hypotheses is provided. Routine referrals are guaranteed within 48 h but are provided most of the times within 24 h. Urgent referrals may be prompted via phone call and are provided within 1 h. One or two first assessments each day are programmed in due advance for patients attending day-hospitals or out-patient clinics.

At the beginning of the working day, a staff meeting for group reading and distribution of received referrals takes place to organize activities. In the afternoon, collegial discussion of clinical cases and supervision of residents is carried out. Psychiatric consultation letters are then written electronically and sent back to referring wards, accompanied by verbal details by phone when necessary. For urgent referrals, the back-referral system is accelerated and often takes place in the ward, right after visiting the patient. Clinical procedures for psychiatric consultations follow indications from international evidence adapted to local context features. An electronic database was adopted by the CLPS for audit purposes after the involvement of the service in the research studies on CLP promoted in Europe by the ECLW in the late ‘90s [25,26,27]. For privacy reasons, it was decided to work on the aggregate data provided by this electronic database (through the request of appropriate “queries”). A complete, homogeneous, and reliable electronic documentation is available from year 2000. In this study, all first psychiatric consultations carried out at the CLPS in the period 2000–2019 (20 years; *N* = 19,278) were considered and analysed. More in detail, the following clinical and non-clinical variables were considered: number of first consultations per year; age (in years) and gender (male/female) of patients; waiting time (time—in days—from referral to actual performance of the assessment); number of referrals per year received by seven top-referring hospital wards (internal medicine, oncology, gastroenterology, headache centre, transplant surgery, general surgery, nephrology-dialysis); number of the eight most recurring reasons for referral among both sub-populations (abuse of alcohol/drugs/psychotropic drugs; anxiety; delirium; depression; medical unexplained physical symptoms; pre-orthotopic liver transplantation (pre-OLTx); psychomotor agitation; re-evaluation of psychopharmacological therapy); proportion of urgent referrals (defined as to be carried out within three hours from request). Scalar and interval variables were described as means and standard deviations (SD), ordinal and categorical variables as absolute values and percentages. Statistically significant differences between OV65 and NOV65 were searched for by means of the t-test for interval and scalar variables and of the Chi-square test for categorical and ordinal ones. The ANOVA test was performed to evaluate the variation of an interval or scalar variable over the years. This research was approved by local Ethics Committee (Prot. AVEN 886/2020/OSS*/AUSLMO).

## 3. Results

Table 1 summarizes a complete description of the sample, including results of statistical analysis of comparison between OV65 and NONOV65 over time. Over the 20 years of activity considered, the Modena CLPS carried out 19,278 consultations: 11,783 (61.12%) for NOV65 and 7495 (38.88%) for OV65. The year with the highest number of consultations performed was 2005 (*N* = 1206; 6.26%), while the one with the lowest was 2001 (704; 3.65%). The total mean annual number of 1st consultations was 963.90 (SD = ±157.93), 589.05 for NOV65 (SD = ±108.06) and 374.65 (SD = ±75.46) for OV65 populations.

Figure 1a,b describe the variation over time in the absolute numbers and proportions of total assessments for OV65 and NONOV65. Both comparison of annual numbers of consultations for OV65 vs. NONOV65 and variation of proportion over time were found statistically significant, with evidence of an increase in the number of referrals for older patients (t = 9.69; df = 19; *p* ˂ 0.001).

The mean age of the entire sample was 57.86 (SD = ±18.09): the variations over years of the mean age of NOV65 (45.93; SD = ±12.29) and OV65 (75.90; SD = ±7.25) were statistically significant (F = 15.59; df = 19; *p* ˂ 0.001 and F = 4.08; df = 19; *p* ˂ 0.001, respectively), though that observed among OV65 was more marked (Figure 2). 

Females were most frequent than males in the general sample (*N* = 10,376; 53.82%) as well as among both the NOV65 subpopulation (*N* = 6373; 54.09%) and the OV65 (4001; 53.38%). The variation of the male/female ratio over the years was statistically significant (Χ^2^ = 134.97; df = 19; *p* ˂ 0.001) (Figure 3), and the same was found among both the subpopulations (NOV65 population: Χ^2^ = 84.11; df = 19; *p* ˂ 0.001 and OV65 population Χ^2^ = 62.31; df = 19; *p* ˂ 0.001). 

The mean waiting time for a consultation was 1.52 days (SD = ±4.47) in the total sample, 1.75 (SD = ±5.09) for NOV65 and 1.17 (SD = ±3.31) for OV65. The variation across the years of the waiting time in the two subpopulations was statistically significant (t = 7.62; df = 14,473; *p* ˂ 0.001) (Figure 4), as well as for each individual sub-population (NOV65: F = 19.86; df = 16; *p* ˂ 0.001 and OV65: F = 12.47; df = 16; *p* ˂ 0.001). 

A peculiar feature of this trend, for both populations, should also be noticed: mean waiting times increased considerably in 2008 and then dropped to values even lower than before from 2013 onwards. This finding is chronologically related to the increase, from 2007, in the collaboration of the CLPS with the outpatient clinic of the liver transplant unit. At first, this required an adjustment process of organization, since patients came from different parts of the national territory and it was more difficult to synchronize all different diagnostic procedures, including the mandatory psychiatric evaluation. The organization gradually improved over time, resulting in mean waiting times even lower than before in more recent years.

The ward that requested the highest number of consultations was internal medicine, both in the whole sample (*N* = 6749; 35.01%) and in the two age-subpopulations, though the rate of referrals from this ward decreased constantly over the 20 years for both adults (aged between 18 and 64 years) and older people (aged 65 years and more). On the contrary, the rate of referrals from the transplant unit and gastroenterology increased over the years. This is the expression of the establishment of a liver transplantation assessment circuit that, as mentioned before, took place in the Modena Hospital in 2007, being these two the wards where potential candidates for liver transplantation are admitted and referred for mandatory psychiatric consultation, according to the clinical protocol. Figure 5 (from (a) to (g)) illustrates trends over time of proportions of referrals from the different wards in the two age groups. The comparison between proportions of referrals in the two age groups across time was statistically significant in all wards but gastroenterology (Χ^2^ = 26.38; df = 17; *p* = 0.07) and nephrology-dialysis (Χ^2^ = 29.83; df = 19; *p* = 0.06). 

The most common reason for referral was clinical suspicion of depression both in the entire sample (*N* = 4591; 23.81%) and in the two age subpopulations, though this was more evident among the OV65 (*N* = 2227; 29.71% vs. *N* = 2364; 20.06%), with a statistically significant difference (Χ^2^ = 45.66; df = 15; *p* ˂ 0.001). Trends over time in the two age subgroups were different in a statistically significant way also when the reasons for referral was clinical suspicion of alcohol/substance/psychotropic drug abuse (Χ^2^ = 25.50; df = 15; *p* = 0.04), medically unexplained physical symptoms (MUPS) (Χ^2^ = 33.42; df = 19; *p* = 0.02), pre-OLTx (Χ^2^ = 53.82; df = 14; *p* ˂ 0.001) and rehearsal of on-going psychotropic medications (Χ^2^ = 26.80; df = 14; *p* = 0.02), but not for delirium, psychomotor agitation and anxiety. Figure 6 (from (a) to (h)) illustrates the trends over time of such proportions.

Urgent referrals were more frequent among NOV65 (*N* = 3070; 26.05%) than in the OV65 subpopulation (*N* = 1619; 21.60%). The trend over time in the proportion of urgent vs. non-urgent referrals was found to be different with statistical significance between the two age groups (Χ^2^ = 428.99; df = 19; *p* ˂ 0.001). Figure 7 illustrates such trend.

## 4. Discussion

The clinical activities of a northern Italy hospital-based CLPS were analysed over a period of 20 years, with the aim to recognize and discuss relevant features and changes related to the population referred for psychiatric consultations, with specific attention to comparison of different features related to age. Such an analysis was felt relevant, and confirmed to be, to provide helpful hints to improve the performances of the CLPS. For example, it could suggest improvements in organizational aspects (e.g., dedicated pathways to care), or training activities (e.g., on psychogeriatric topics), or further research initiatives.

Activities dedicated to the older population represent a significant amount of the whole of clinical responsibilities of the CLPS, an average of about 40%. Moreover, this amount progressively increased over the 20 years considered, and the variation of mean age was more evident among older patients. All these findings are consistent with the phenomenon of aging of both general and hospital populations [13,20,28] and confirms the relevance to study these patterns of utilization. Both consultees and consultants should be aware of specific needs of their geriatric patients in terms of mental health when they refer and assess them, respectively: these needs are known to be different than those of younger patients [14,16,29], and require specific knowledge and skills (i.e., the ability to perform a differential diagnosis between depression and hypoactive delirium, or a safe and appropriate management of psychotropic medications). 

This was also clearly outlined by results related to most common reasons for referrals: depression, the most common reason for psychiatric referral both in the total sample and in the two age-groups, was particularly frequent in the over-65 patients. 

Depression in older patients is easily missed and mismanaged, due for example to multimorbidity and difficulties in appropriate interpretation of somatic symptoms or to uneasiness of clinicians in using antidepressants for fear of side effects or interactions [30,31], but also to a more implicit conceptual reason, that is the tendency to feel that it is normal or inevitable to be depressed when old [30,32]. The increase here reported may be a positive sign, meaning that clinicians are more aware of the need to properly address and treat depression among their older patients. No statistically significant change in the proportion of diagnoses of delirium in the two age groups was found, this being also a confirmation not only of the epidemiology of this condition, but also of the good reliability of physicians in diagnosing it. 

The amount of referrals received changed significantly in the two subpopulations for other diagnostic categories: abuse of alcohol, substances or medications was more common but also much more variable among younger patients. Referrals for pre-transplant assessments were also more common among patients with less than 65 years, as expected, being age a significant parameter that influences the choice of candidates by surgeons, though the number of referrals with this indication evidently increased in both age groups in recent years, consistently with the growing importance as surgical hub for transplantation of the Modena centre [33]. Generic referrals for psychomotor agitation, increasing particularly for older patients until 2012, have become less common in more recent years: this may also reflect an improvement in the diagnostic skills of consultees, who are able to use more detailed diagnostic frameworks such as that of delirium, as already commented. The observation of such changes, allowed by the long-time of activity here considered, is also an indirect evidence that day-by-day work on liaison and more formal training initiatives promoted by the CLPS are effective in the long run and help a better use of resources [34]. Another relevant change observed over the 20 years in both age-groups was the drop in the referrals due to MUPS: this finding is maybe the expression of a conceptual twist in the nosography of somatisation and the so-called functional disorders, variously addressed in the scientific literature [35]. 

Female patients referred to the CLPS were more common both among older and among younger patients, but also their proportion progressively decreased along the years in both subpopulations. This change, though, was more marked in the older research participants. One possible way to interpret this finding is the increased amount of patients referred for pre-OLTx, who are mostly male [36]. Gender-related issues in geriatric patients are very relevant, considering the higher proportion of more severe forms of some disorders in male patients, e.g., depression [37]: awareness on the consequent increased risk of self-harm behaviours should be an important priority in the organization of the CLPS as well as in the definition of training activities.

The Modena CLPS confirms to perform well in terms of waiting times for back-referral, with an average of 1 day and half. This time is even less for older patients, who may be felt as a priority in the scheduling of the activities of the day by the consultants. Urgent referrals, though, were more common for younger subjects, but decreased considerably in both groups over the 20 years, as a possible sign that consultees have learnt to “trust” the CLPS in providing prompt reaction when needed. With reference to changes in the trend of waiting times, moreover, a significant increase was noticed, in both populations, around 2008, followed by a big drop in 2013. This trend is related to the history of the Modena CLPS, which, in 2008, increased significantly its regular collaboration with the outpatient clinic of the Liver Transplant Unit, resulting, for some years, in more difficult and timely organization of appointments. This finding is a clear example of how useful the regular recording of clinical activities is in understanding trends and changes that reflect internal and external events [3].

The study has some limitations that must be clarified to understand the discussion of the results: (1) the professionals who collected the data were different over the years. Each professional was given a short introductory course however this may not have prevented some errors in entering data; (2) data collection was done following privacy regulations but not through a dedicated software. Using a spreadsheet did not allow to avoid errors in data entry; (3) for the majority of hospital wards, requests for psychiatric consultation were not defined by a protocol or a criteria check-lists: medical doctors decided in total autonomy when to request a consultation for their hospitalized patient. This subjectivity may have characterized the distribution of requests in the different wards; (4) some wards (for example transplant surgery) had a very strong collaboration with the CLPS, using its support in their protocols. This strong synergy has certainly influenced the number of requests from that wards; (5) the database does not collect variables measuring level of dependence, activities of daily living, social support or even degree of frailty, whose possible interesting changes over time we were therefore not able to document.

## 5. Conclusions

One major goal of CLPSs operating in the GH should be the acquisition of an attitude more strongly focused on prevention and health promotion. The epidemiologic approach to clinical data achieved by this study, together with its wide time-span (20 years of activity), supports the recognition of obstacles, opportunities and other needs in the organization of the CLPS, in order to improve its effectiveness in dealing with medical-psychiatric comorbidity of older patients.

Consultations for older patients accounted for a large amount, more than one third, of the whole activities of the CLPS, and a significant increasing trend was evident. Depression was by far the most common diagnosis not only in the general population assessed by the CLPS, but in the older age group particularly. Changes in mean lag-times between referral and assessment as well as in proportion of urgent/routine referrals were evident.

These findings suggest that along with the phenomenon of progressive ageing in hospital populations, there is an increased need for hospital-based CLPS to potentiate their skills in managing old-age specific psychopathology and tailor their organization, training, and research activities accordingly. The present study provides support to a better tuning to the needs of older people by identifying high-risk disease patterns or promoting dedicated clinical processes (i.e., detection of mild cognitive impairment, coordination with primary care or social services, prevention of delirium).

## Figures and Tables

**Figure 1 ijerph-17-07389-f001:**
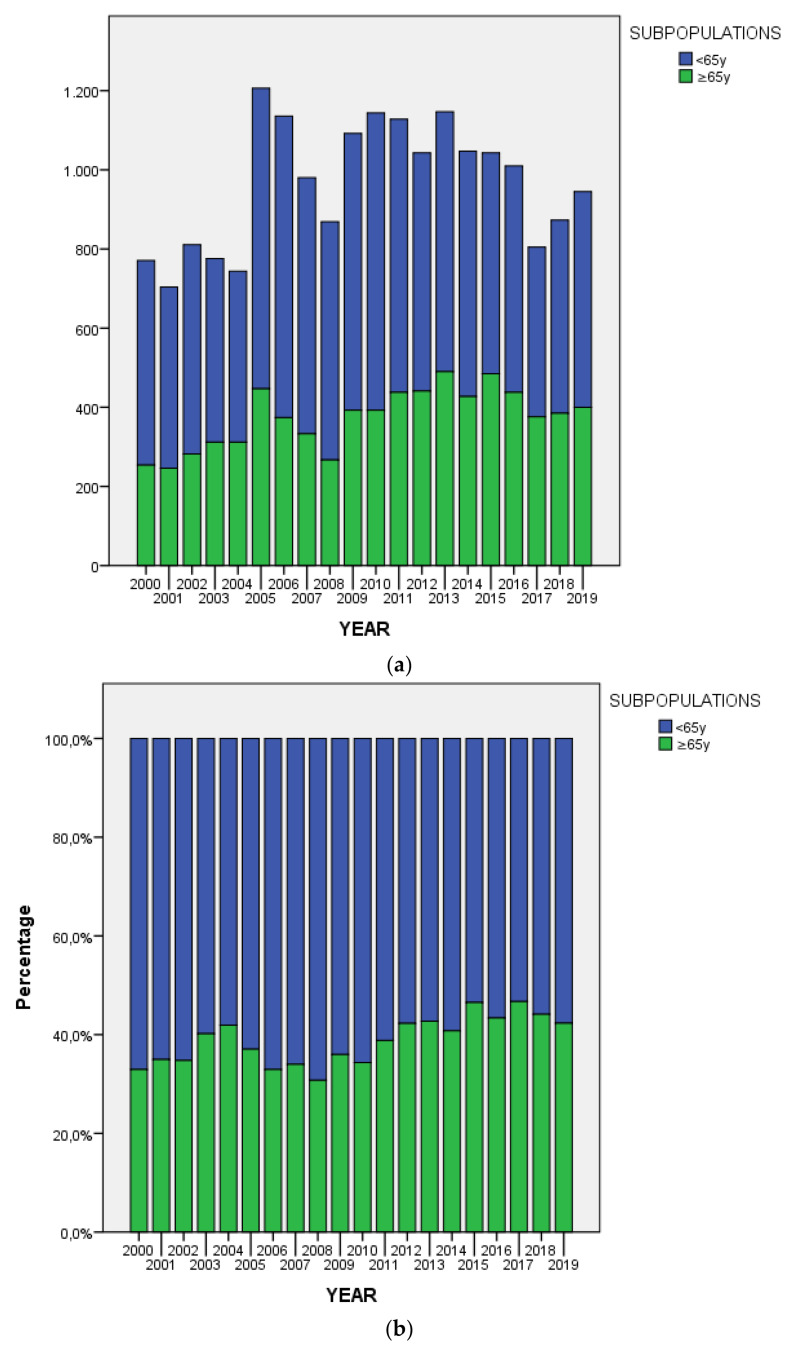
Consultations carried out from 2000 to 2019, divided into the two subpopulations. (**a**) Absolute value of psychiatric consultations carried out; (**b**) Percentage of psychiatric consultations carried out.

**Figure 2 ijerph-17-07389-f002:**
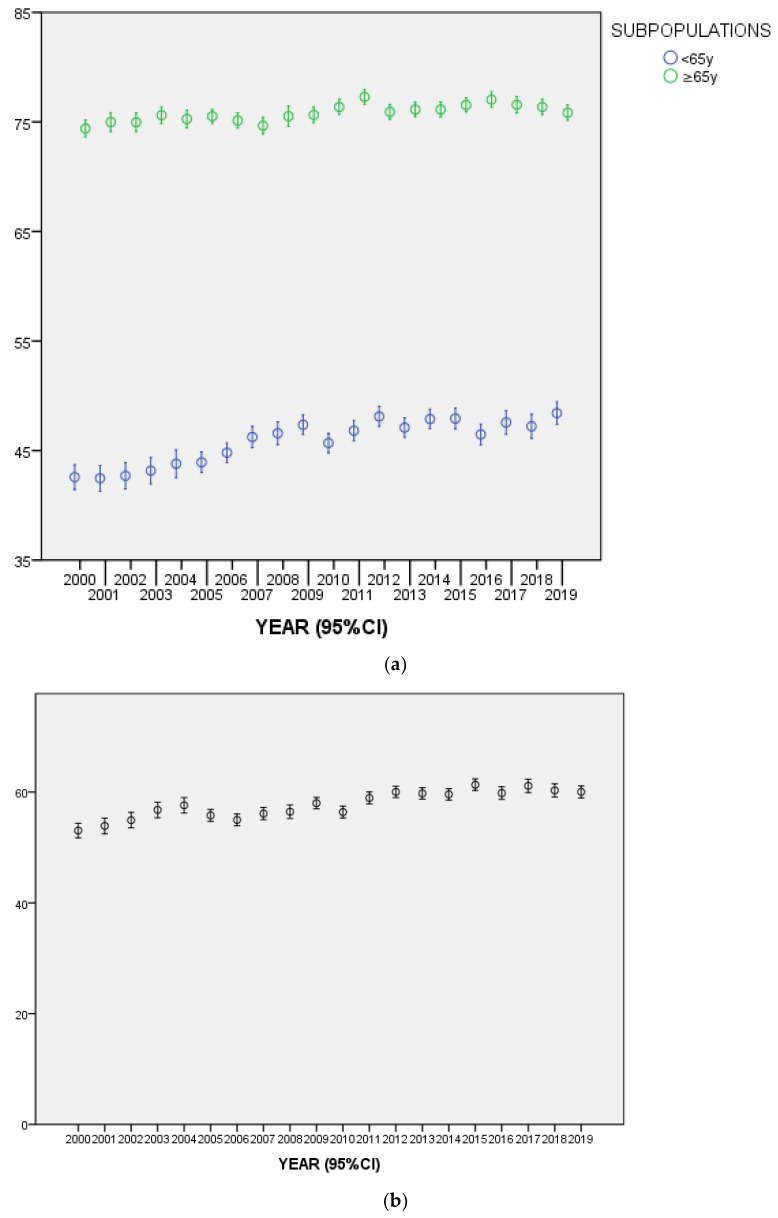
(**a**) Mean age of the two subpopulations (NOV65 and OV65). (**b**) Mean age of the entire sample.

**Figure 3 ijerph-17-07389-f003:**
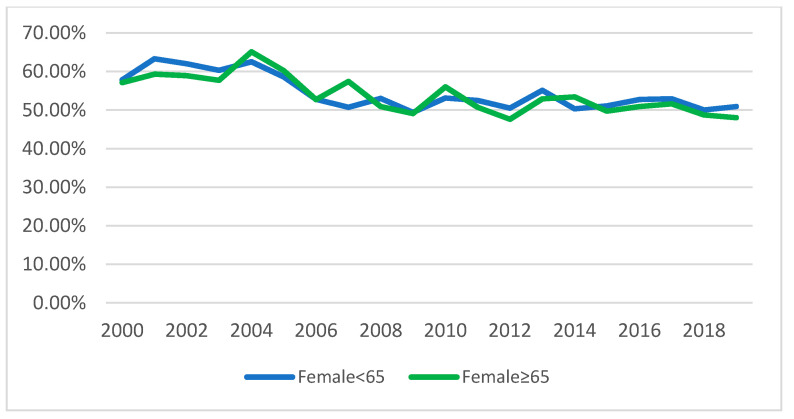
Percentage of consultations, per year, made to female patients.

**Figure 4 ijerph-17-07389-f004:**
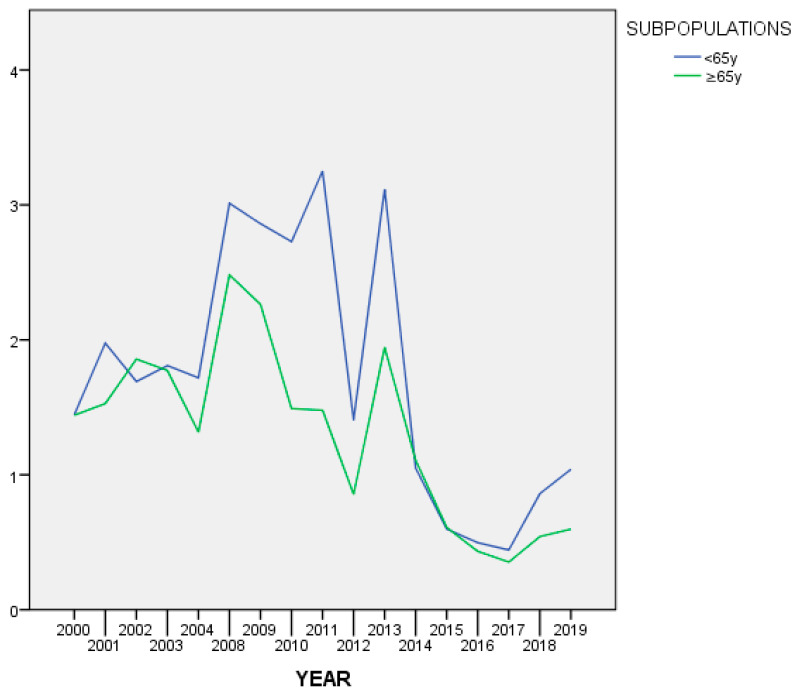
Mean waiting time from referral to assessment (days).

**Figure 5 ijerph-17-07389-f005:**
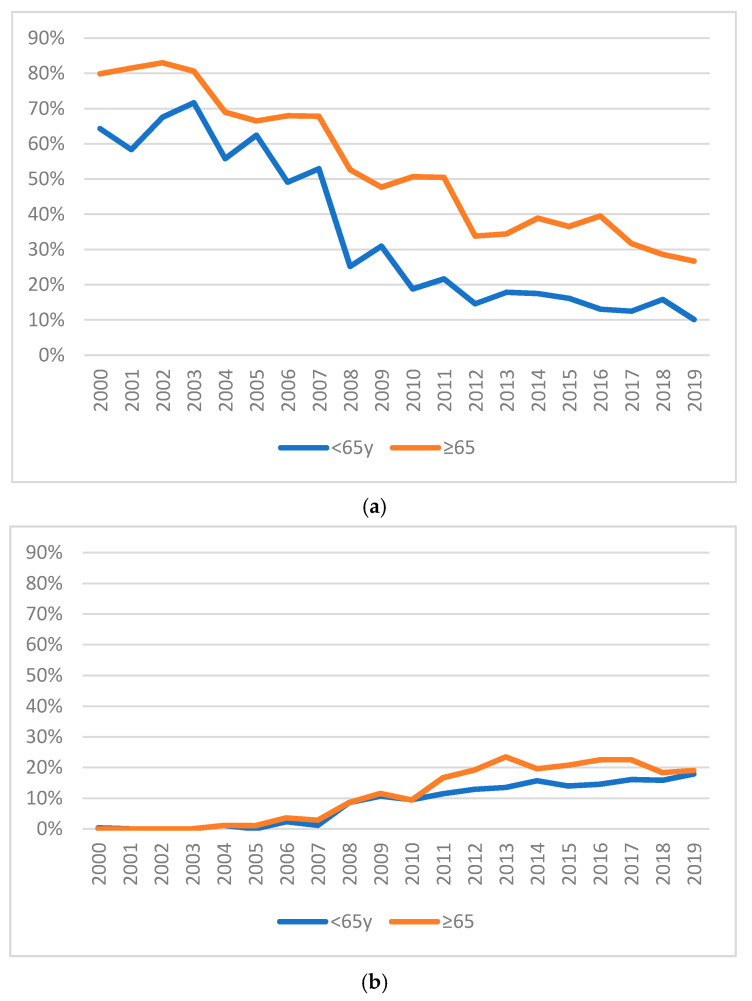
Trends over time of percentage of referrals from wards for the subgroups NOV65 and OV65. (**a**) Internal medicine; (**b**) Oncology; (**c**) Gastroenterology; (**d**) Headache centre; (**e**) Transplant surgery; (**f**) Surgery; (**g**) Nephrology-dialysis.

**Figure 6 ijerph-17-07389-f006:**
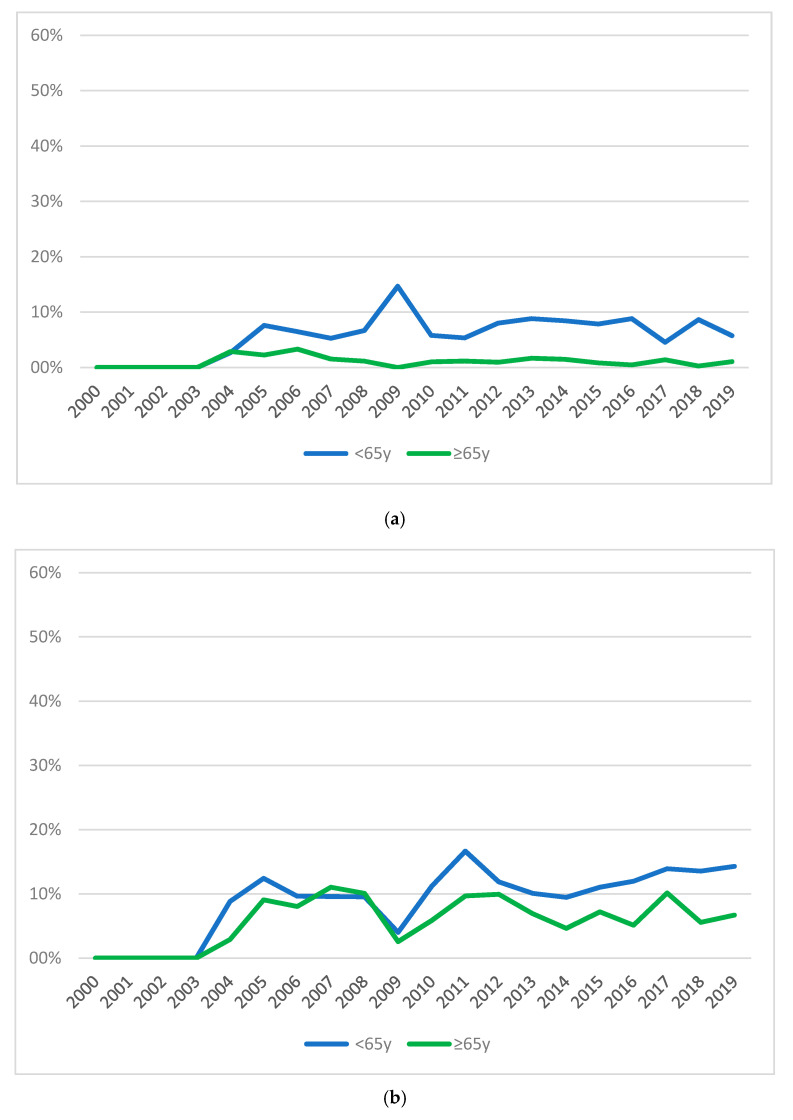
Trends over time of percentage of reasons for referrals for the subgroups NOV65 and OV65. (**a**) Abuse of alcohol/drugs/psychotropic drugs; (**b**) Anxiety; (**c**) Delirium; (**d**) Depression; (**e**) Medical Unexplained Physical Symptoms; (**f**) Pre-OLTx; (**g**) Psychomotor agitation; (**h**) Re-evaluation of psychopharmacological therapy.

**Figure 7 ijerph-17-07389-f007:**
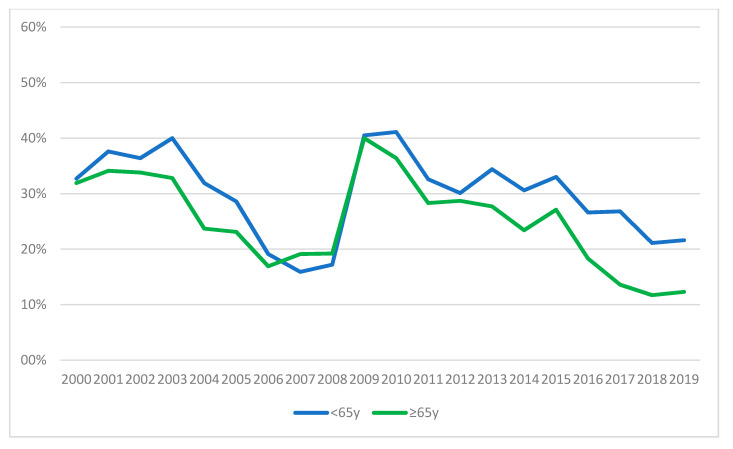
Percentage of urgent requests over time.

**Table 1 ijerph-17-07389-t001:** Descriptive analysis of the collected clinical and sociodemographic variables.

	Entire Population (*N* = 19278)	Not Over 65 Subpopulations(NOV65)	Over 65 Subpopulations(OV65)	Not Over 65 (NOV65) vs. Over 65 (OV65) Subpopulations during Years	Variation of Mean/Ratio during Years
	Mean	DS	Mean	DS	Mean	DS	NOV65	OV65
**Number of consultations**	963.90	±157.93	589.05	±108.06	374.65	±75.46	t = 9.69; df = 19; *p* ˂ 0.001	t = 24.38; df = 19; *p* ˂ 0.001	t = 22.21; df = 19; *p* ˂ 0.001
**Age**	57.86	±18.09	45.93	±12.29	75.90	±7.25	Not calculated	F = 15.59; df = 19; *p* ˂ 0.001	F = 4.08; df = 19; *p* ˂ 0.001
**Mean waiting time from referral to assessment (days)**	1.52	±4.47	1.75	±5.09	1.17	±3.31	t = 7.62; df = 14473; *p* ˂ 0.001	F = 19.86; df = 16; *p* ˂ 0.001	F = 12.47; df = 16; *p* ˂ 0.001
	*N*	%	*N*	%	*N*	%	
**Sex**								
Male	8885	46.09%	5394	45.78%	3489	46.55%	Χ^2^ = 134.97; df = 19; *p* ˂ 0.001	Χ^2^ = 84.11; df = 19; *p* ˂ 0.001	Χ^2^ = 62.31; df = 19; *p* ˂ 0.001
Female	10,376	53.82%	6373	54.08%	4001	53.38%
Missing	17	0.09%	16	0.14%	5	0.07%		
**Number of referrals according to ward**							
Internal medicine	6749	35.01%	3411	28.95%	3338	44.54%	Χ^2^ = 297.45; df = 19; *p* ˂ 0.001	Not calculated
Oncology	1840	9.54%	950	8.06%	890	11.87%	Χ^2^ = 43.51; df = 16; *p* ˂ 0.001
Gastroenterology	1088	5.64%	799	6.78%	288	3.84%	Χ^2^ = 26.38; df = 17; *p* = 0.07
Headache centre	945	4.90%	870	7.38%	75	1.00%	Χ^2^ = 32.84; df = 19; *p* = 0.03
Transplant surgery	979	5.08%	798	6.77%	179	2.39%	Χ^2^ = 44.90; df = 11; *p* ˂ 0.001
Surgery	902	4.68%	508	4.31%	394	5.26%	Χ^2^ = 175.45; df = 19; *p* ˂ 0.001
Nephrology-dialysis	624	3.24%	265	2.25%	359	4.79%	Χ^2^ = 29.83; df = 19; *p* = 0.06
**Reason for referral**								
Abuse of alcohol/drugs/psychotropic drugs	665	3.45%	587	4.48%	78	1.04%	Χ^2^ = 25.50; df = 15; *p* = 0.04	Not calculated
Anxiety	1414	7.33%	981	8.33%	432	5.76%	Χ^2^ = 23.76; df = 15; *p* = 0.07
Delirium	449	2.33%	93	0.79%	356	4.75%	Χ^2^ = 16.94; df = 15; *p* = 0.32
Depression	4591	23.81%	2364	20.06%	2227	29.71%	Χ^2^ = 45.66; df = 15; *p* ˂ 0.001
Medical Unexplained Physical Symptoms	1246	6.46%	93	0.79%	312	4.16%	Χ^2^ = 33.42; df = 19; *p* = 0.02	
Pre-OLTx	1160	6.02%	1043	8.85	116	1.55%	Χ^2^ = 53.82; df = 14; *p* ˂ 0.001
Psychomotor agitation	1581	8.02%	512	4.35%	1068	14.25%	Χ^2^ = 18.29; df = 15; *p* = 0.25
Re-evaluation of psychopharmacological therapy	831	4.31%	481	4.08%	350	4.67%	Χ^2^ = 26.80; df = 14; *p* = 0.02
**Number of urgent referrals**								
Urgent	4689	24.32%	3070	26.05%	1619	21.60%	Χ^2^ = 428.99; df = 19; *p* ˂ 0.001	Χ^2^ = 248.38; df = 19; *p* ˂ 0.001
Not urgent	12,210	63.34%	7178	60.92%	5032	67.14%
Missing information	2379	12.34%	1535	13.03%	844	11.26%

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
