# Peer review of "Is Consultation-Liaison Psychiatry ‘Getting Old’? How Psychiatry Referrals in the General Hospital Have Changed over 20 Years"

_ijerph, 2020, doi:10.3390/ijerph17207389_

Round 1

Reviewer 1 Report

This an interesting, weel-written paper. However, some issues have to be disccussed:

a) Authors have to clear describe the ethical approval

b) The aim of this paper was describing changes of socio-demographic and clinical characteristics in 84 the population aged 65 and over referred to a hospital-based CLPS over a 20-year (2000 - 2019) lag-85 time. The results refer mainly to changes at the clinical and pathological level. In this population it would be interesting to know also changes at the level of dependence, activities of daily living, social support or even degree of frailty.

Author Response

We are grateful to reviewer 1 who brought very important and accurate comments and suggestions to our attention.

1)    Authors have to clear describe the ethical approval

The manuscript reports in aggregated form routine clinical data collected for audit purposes. However, as agreed with the assigned Editor, we proceeded to lodge the request for formal approval to the local ethics committee. The article will not be published without this approval (expected approximately mid-September).

2)    The aim of this paper was describing changes of socio-demographic and clinical characteristics in the population aged 65 and over referred to a hospital-based CLPS over a 20-year (2000 - 2019) lag-time. The results refer mainly to changes at the clinical and pathological level. In this population it would be interesting to know also changes at the level of dependence, activities of daily living, social support or even degree of frailty.

We agree with the reviewer: it would be indeed of interest to also know about such relevant information about functioning. Unfortunately, these variables are not routinely collected by the CLPS, being more specific for elderly populations, whereas the CLPS operates over a wider range of age groups, so this is a fixed limitation which we have acknowledged in the limitations section.

Page 21, Lines 268-370

5) the database does not collect variables measuring level of dependence, activities of daily living, social support or even degree of frailty, whose possible interesting changes over time we were therefore not able to document

Reviewer 2 Report

  1. In the Methods, some description of how the data was extracted from the IT system is needed, as this would affect the data quality. Was ethical approval needed? Have all or most the CLPS consultation data been entered to the electronic system since 2000?

  1. The data actually included both older (OV65) and younger adults (NOV65). The results of the younger adults are important as well. They accounted for nearly two thirds of the consultations. I suggest the study title and objective not to limit to older patients as most of the analyses aimed to compare the two groups on various useful parameters. In fact, ageing of the population is a global phenomenon. This would be reflected in consultations in other specialties as well.

  1. Comparison of the mean age between the two groups seems problematic. The age of the NOV65 group is capped at 64. It may be more meaningful to show the change in mean age of the entire sample over the years. Besides, the change seems more marked for the NOV65, not OV65 as stated.

  1. Instead of a general wording of “variations”, the main text should describe the rise in waiting time from referral to assessment around 2008-2013, followed by a huge drop in it. This can guide the readers to understand the trend.

  1. Figure 6 shows that huge drop of medical unexplained physical symptoms with the rise of other diagnosis such as anxiety and depression. The link between these trends should be suggested and interpreted, probably following the point about somatic symptoms in the discussion section. Currently, the analysis focuses on comparing the NOV65 and OV65 but ignores other interesting trends observed.

  1. Many formatting issues. The positions of the figure titles are very confusing. It is often unclear whether the title belongs to the current figure or the next one as they stick together. Percentages (%) should be stated in the reasons for referrals for the OV65 in Table 1. The y-axis units for figures 5 -7 should be more standardized.

  1. I think the findings are meaningful but how they would have an impact to local or international development of CLPS is quite vague. The conclusions should be more closely based on the findings of this paper.

  1. Typo, “NONOV65” in the Abstract and the beginning of the Results.

Round 2

Reviewer 1 Report

The authors introduced some changes in the text, which are welcome. However, a central aspect of any study is questions of an ethical nature. The authors do not refer to the way in which they obtained permission to carry out the study or the opinion of the Ethics Committee (e.g. Ethical Ref. nº) . Other suggestions made before did not have the necessary acceptance.

Author Response

you can find in attachment the revised paper with the approval of the Ethics Committee.

Reviewer 2 Report

The authors have responded to the reviewer's questions well. This version is fine for publication. As the authors mentioned, they would wait for the formal ethics approval. 

Author Response

(The authors gave the same response as above.)
